# Allochthonous *Trichoderma* Isolates Boost *Atractylodes lancea* Herb Quality at the Cost of Rhizome Growth

**DOI:** 10.3390/jof10050351

**Published:** 2024-05-14

**Authors:** Kuo Li, Huaibin Lin, Xiuzhi Guo, Sheng Wang, Hongyang Wang, Tielin Wang, Zheng Peng, Yuefeng Wang, Lanping Guo

**Affiliations:** 1School of Chinese Materia Medica, Guangdong Pharmaceutical University, Guangzhou 510006, China; lk990315@163.com (K.L.); linhb2662@163.com (H.L.); 2State Key Laboratory for Quality Ensurance and Sustainable Use of Dao-di Herbs, National Resource Center for Chinese Materia Medica, China Academy of Chinese Medical Sciences, Beijing 100700, China; guoxiuzhi@163.com (X.G.); mmcniu@163.com (S.W.); wanghybio@163.com (H.W.); wtl82@163.com (T.W.); zzpengzheng@sina.com (Z.P.); 3Key Laboratory of Biology and Cultivation of Herb Medicine, Ministry of Agriculture and Rural Affairs, Beijing 100700, China

**Keywords:** *Atractylodes lancea*, *Trichoderma harzianum*, root rot resistance, medicinal compound accumulation, plant–microbe selectivity

## Abstract

*Atractylodes lancea* is a perennial herb whose rhizome (AR) is a valuable traditional Chinese medicine with immense market demand. The cultivation of *Atractylodes lancea* faces outbreaks of root rot and deterioration in herb quality due to complex causes. Here, we investigated the effects of *Trichoderma* spp., well-known biocontrol agents and plant-growth-promoters, on ARs. We isolated *Trichoderma* strains from healthy ARs collected in different habitats and selected three *T*. *harzianum* strains (Th2, Th3 and Th4) with the strongest antagonizing effects on root rot pathogens (*Fusarium* spp.). We inoculated geo-authentic *A*. *lancea* plantlets with Th2, Th3 and Th4 and measured the biomass and quality of 70-day-old ARs. Th2 and Th3 promoted root rot resistance of *A*. *lancea*. Th2, Th3 and Th4 all boosted AR quality: the concentration of the four major medicinal compounds in ARs (atractylon, atractylodin, hinesol and β-eudesmol) each increased 1.6- to 18.2-fold. Meanwhile, however, the yield of ARs decreased by 0.58- to 0.27-fold. Overall, Th3 dramatically increased the quality of ARs at a relatively low cost, namely lower yield, showing great potential for practical application. Our results showed selectivity between *A. lancea* and allochthonous *Trichoderma* isolates, indicating the importance of selecting specific microbial patches for herb cultivation.

## 1. Introduction

*Atractylodes lancea* is a perennial herb and the main source of *Atractylodis* Rhizoma, the dried rhizome of *A. lancea* and traditional Chinese materia medica. The rhizome is pungent, bitter in taste, warm in nature, and belongs to the spleen, stomach, and liver meridians. It dries out dampness, strengthens the spleen, drives away wind and cold, and improves eyesight [1]. In traditional Chinese medicine (TCM), it is a commonly used medicinal ingredient in the treatment of “novel coronavirus pneumonia” [2]. Being an effective medicine for preventing epidemics in China since ancient times, the method of air disinfection by fumigation with *A. lancea* can also effectively combat the spread of epidemics in modern times [3]. In short, *A. lancea* is a widely used Chinese herbal medicine that currently has a large global market.

With the vigorous development of China’s comprehensive health industry and the gradual improvement in global awareness and recognition of TCM, the market demand for Chinese medicinal materials is increasing. Cultivation has become the major source of these Chinese medicinal materials [4]. The *A. lancea* cultivation industry in China has developed for nearly 20 years, with the cultivation area being widespread and increasing every year [5,6]. However, the yield and quality of cultivated *A. lancea* are severely threatened by various factors. For example, the geographical authenticity of the germplasm, the choice of cultivation methods/systems and outbreak of plant diseases in *A. lancea* farms. In particular, root rot of *A. lancea* caused by *Fusarium* species is a soil-borne disease that severely threatens the cultivation and production of *A. lancea* and for which solutions need to be found urgently. Notably, the use of pesticides in the farming of TCM raw products is strictly restricted due to the low contamination standards prescribed by the China Pharmacopoeia (CP). Hence, among the many methods available for controlling plant diseases, the most environmentally friendly method, biological control, seems to be the most suitable and promising for *A. lancea* farming.

*Trichoderma* spp. are some of the most potent fungal antagonists for the biocontrol of plant diseases. They can form symbiotic relationships with plants to induce plant resistance to fungal diseases, promote plant growth and increase plant resilience against abiotic stresses [7,8]. *Trichoderma* spp. have been extensively used to prevent, contain and control plant diseases in agriculture and horticulture, effectively reducing losses caused by plant diseases [9,10,11,12]. *Trichoderma* spp. have a well-established reputation for treating soil-borne diseases such as root rot of cucumber, tomato, peanut and other crops [13,14,15,16,17,18,19,20,21].

There are also studies showing that *Trichoderma* spp. are effective against root rot pathogens of medicinal plants. For example, Ding et al. [22] reported that *T. harzianum* had an obvious biocontrol effect on the *sclerotium* root rot of *Glehnia littoralis* both in the greenhouse and in the field. This *T. harzianum* inoculum could also control the root rot of *Astragalus membranaceus* in the field and protect American *ginseng* (*Panax quinquefolium*) seedlings from damping-off by 60% in the field. In a field experiment conducted by Li et al. [23] in 2015, *Trichoderma* strains significantly promoted seedling survival rate and both fresh weight and dry weight yield of 1-year-old and 2-year-old *Panax notoginseng*. Hu et al. [24] found that *Trichoderma* had a significant inhibitory effect on the pathogen *Fusarium solani* isolated from the rhizosphere of *Ligusticum chuanxiong* in dual-culture experiments. Qiang et al. [25] isolated *Trichoderma* strains from the rhizosphere of *ginseng*; dual-culture experiments showed that the *Trichoderma* isolates had evident inhibitory effects on *Panax ginseng* root rot pathogens.

Reports on the application of *Trichoderma* spp. in the farming of Chinese materia medica have steadily increased in recent years [26,27,28], but most research has only reported improvements in seedling survival and herb yield. Few reports have addressed the effects of *Trichoderma* on the accumulation of bioactive medicinal compounds in the inoculated herbs, reflecting a paucity of attention on herb quality. Furthermore, the majority of these reports based their observation of the biocontrol effects of *Trichoderma* spp. on dual-culture experiments, which are not utterly sufficient. Plant inoculation experiments or/and field trials should always be the most convincing.

Aiming at the biocontrol of root rot disease in *A. lancea* farming, in this study, we isolated *Trichoderma* strains from the rhizosphere of healthy wild *A. lancea* collected in Liaoning Province and healthy farmed *A. lancea* collected in Inner Mongolia Autonomous Region, China. We screened for *Trichoderma* isolates that exhibited strong biocontrol potential against root rot pathogens and investigated the function of these *Trichoderma* strains when used as an inoculum for *A. lancea* plantlets. We selected three *Trichoderma* isolates with the strongest biocontrol potentials. In our inoculation experiments, they all exhibited biocontrol functions against root rot disease and promoting effects on the growth and medicinal compound accumulation to different extents. Therefore, we subsequently investigated the optimum culturing and sporulation conditions for these three *Trichoderma* strains as preparation for their future application in *A. lancea* farming.

## 2. Materials and Methods

### 2.1. Plant and Fungal Materials

*A. lancea* plantlets used for the inoculation experiments were acquired and propagated via tissue culturing following the protocols of Wang et al. [29]. Briefly, first, plantlets were proliferated using the tillering medium (Appendix A); subsequently, tillered plantlets were separated and cultured using the rooting medium (Appendix A), eventually forming plantlets with adventitious roots ready for inoculation and transplantation into pot soil. The propagated plantlets and the plants in the pot soil experiments were all cultured in growth chambers (Saifu Co., Ltd., Ningbo, China) at 26 °C, with 65% relative humidity and a 12 h (h)/12 h light–dark cycle.

Root rot pathogen strains, namely a *Fusarium solani* strain (with the ID ‘NJ5′) and an *F. oxysporum* strain (with the ID ‘NJ40′; referred to as Fs and Fo, respectively, hereafter) were kindly provided by Associate Professor Tielin Wang of the Chinese Academy of Chinese Medical Sciences. Both were pathogens of root rot disease and isolated from the necrotic rhizome tissue of farmed *A. lancea* with root rot symptoms. Pathogenicity had been verified by T.W. [30].

Healthy wild *A. lancea* samples were collected in Liaoyang City, Liaoning Province, China (123°39′42″ E, 41°4′17″ N). Healthy farmed *A. lancea* samples were collected in Chifeng city, Inner Mongolia Autonomous Region, China (118°29′55″ E, 42°4′52″ N). Plants were carefully collected with intact underground compartments attached to the rhizosphere soil and some associated bulk soil. Plants were placed on ice after collection and immediately transported to the lab for *Trichoderma* isolation.

### 2.2. Isolation, Identification and Culturing of Fungi

The culturing medium used in this study was all prepared following the Appendix A. RBA medium (Beijing Land Bridge Technology Co., Ltd., Beijing, China) in petri dishes was applied for the isolation of *Trichoderma* strains. Isolation was performed following Wang et al. [31]. Soil that adhered to the underground compartment of the fresh *A. lancea* samples we collected was carefully removed and gathered as the soil sample for *Trichoderma* isolation. Then, we separated the rhizome and the fibrous roots using a scalpel, and separately placed them and the soil sample into three autoclaved flasks containing sufficient sterile water to immerse the samples. The flasks were then sealed and oscillated at 200 rpm, 25 °C for 2 h to generate a rich suspension of the soil or rhizosphere microbes of the samples. The suspension of each sample was then diluted by 1×, 10×, 50× and 100× gradients and evenly spread on RBA medium, 100 μL on each plate, three replicates for each gradient, under sterile conditions.

The RBA plates were then placed in an incubator at 28 °C in the dark and observed daily by the naked eye. After 48 hours, mycelia that emerged were carefully picked up at the edge and inoculated onto PDA (Oxoid Co., Ltd., Basingstoke, UK) plates under sterile conditions to acquire pure cultures of novel isolates, then incubated at 28 °C in the dark [29,30]. The morphology of the novel isolates was observed every day, and only those that exhibited characteristics of *Trichoderma* spp. (fast colony growth, fluffy mycelium structure, gradual color change of colony to yellow, green, then dark green) were subjected to further culturing and identification.

The identification of potential Trichoderma isolates was performed via microscopic observation and sequencing of the ITS (internally transcribed spacer) [32] sequence using the 1TS1 5′-TCCGTAGGTGAACCTGCGG-3′ and ITS4 5′-TCCTCCGCTTATTGA- TATGC-3′ primer pair. Further, for more reliable identification, we performed sequencing of the TEF1 and RPB2 sequences for the isolates that were identified as *Trichoderma* by ITS using the EF1-728F 5′-CATCGAGAAGTTCGAGAAGG-3′ and TEF1LLErev 5′-AACTTGCAGGCAATGTGG-3′ primer pair, and the fRPB2-5f 5′-GA(T/C)GA(T/C)(A/C)G(A/T)GATCA(T/C)TT(T/C)GG-3′ and fRPB2-7cr 5′-CCCAT(A/G)GCTTG(T/C)TT(A/G)CCCAT-3′ primer pair [33]. Sequencing was performed by Ruibo Xingke Biotechnology Co., Ltd. (Beijing, China). The ITS sequences were then searched on NCBI (https://www.ncbi.nlm.nih.gov/, accessed on 14 January 2024) via BLAST. Molecular identification was thus performed via ITS sequence alignment.

Fungal vouchers and the pathogens (Fo and Fs) were cryogenically preserved in 5 mm diameter round pieces of PDA culture immersed in 50% (*v*/*v*) glycerol at −80 °C at the Chinese Academy of Chinese Medical Sciences. When put to use, cryogenically preserved fungal inoculum was firstly cultured on PDA at 28 °C for 7 days, and then the fresh culture was used as inoculum for further experiments. All fungi were cultured on PDA at 28 °C in the dark. Liquid culture was performed using PDB (Solarbio Co., Ltd., Beijing, China) at 28 °C, 200 rpm. Liquid *Trichoderma* or *Fusarium* inoculants were cultured for 5 days and 7 days, respectively.

### 2.3. Dual-Culture Experiments

The biocontrol potentials of the *Trichoderma* isolates against Fo and Fs were evaluated using the dual-culture (DC) method on PDA in 9 cm diameter petri dishes, following the method of Wang et al. [31]. Briefly, fungal inoculums were placed on PDA with a proper distance from each other, then cultured and observed for 20 days. Plates inoculated with *Fusarium* alone were used as controls (CK).

Meanwhile, dual-culture experiments following the method of Ma et al. were performed to test the inhibitory effects of the volatile organic compounds of our *Trichoderma* isolates (Figure 1E) [34]. The tested Trichoderma and Fusarium strains were inoculated on PDA simultaneously, and then sealed jointed to be cultured in the same close space. An Fo or Fs plate was taken as the control (CK), with a blank PDA plate used as the control. After 3 days of culture, the colony diameter D was measured.
Inhibitory rate (IR, %) = (D_CK_ − D_DC_)/D_CK_ × 100%

### 2.4. Inoculation of A. lancea Seedlings

*A. lancea* plantlets harvested from the rooting medium were carefully washed with distilled water to remove the solid medium. The plantlets were then placed in sterile tissue-culture bottles with the root immersed in sterile water and subjected to a 48 h hardening process in the growth cabinet. Subsequently, the hardened plantlets were subjected to inoculation.

Liquid fungal culture in PDB was used as an inoculum. Liquid Fo and Fs cultures were mixed to form the pathogen inoculum, with the concentration of Fo and Fs being 2.5 × 10^7^ cfu/mL each. The fungal concentrations of the liquid Th2, Th3 and Th4 cultures were 6.0 × 10^6^ cfu/mL, 1.0 × 10^7^ cfu/mL and 1.0 × 10^7^ cfu/mL, respectively. They were all adjusted to 6.0 × 10^6^ cfu/mL using sterile water and subsequently used as the *Trichoderma* inocula. Inoculation was performed by placing the hardened *A. lancea* plantlets in tissue-culture bottles, with the root immersed in inoculum, and then placed in the growth cabinet. Inoculated plantlets were then planted in sterile peat soil in 150 mL cups (one cup for each plantlet) and cultivated in the growth chamber with sufficient watering.

Plantlets of the CK group were mock-inoculated with sterile water for 24 h. The Fo + Fs group: inoculated with the pathogen inoculum for 24 h. The Th2/Th3/Th4 group: inoculated with the corresponding *Trichoderma* inoculum for 24 h. The Fo + Fs_T groups: treated with the pathogen inoculum for 24 h, then, respectively, treated with the corresponding *Trichoderma* inocula for 24 h; the T_Fo + Fs groups: treated with the *Trichoderma* inocula and then the pathogen inoculum for 24 h each.

### 2.5. Measurement of A. lancea Biomass and Content of Medicinal Compounds

After 70 days of growth, the *A. lancea* plants were harvested. Harvested *A. lancea* plants were carefully washed with tap water to remove the soil. Clean *A. lancea* plants were then gently wiped to remove water and then weighed. Weighed samples were temporarily preserved in liquid nitrogen and then preserved at −80 °C for further experiments. Dry weight data were measured after freeze-drying.

Measurement of hinesol, β-eudesmol, atractylon and atractylodin content was performed following the exact protocols and the standard curve of Wang et al. [29]. The *A. lancea* seedlings used in this study were from Jiangsu Province. Therefore, the dry rhizomes were rich in atractylodin and atractylon but only contained a scarce amount of hinesol and β-eudesmol.

### 2.6. Studies on the Optimal Culturing and Sporulation Conditions for the Trichoderma Isolates

Our selected *Trichoderma* isolates were cultured on different media (Appendix A) in order to screen for the optimal growth and sporulation medium (28 °C, dark). Optimal light/dark conditions (24 h continuous light, 24 h continuous dark, and light and dark alternate 12 h/12 h), temperature (15, 20, 25, 30, 35 and 40 °C), pH (5, 6, 7, 8, 9, 10) for the colony growth and sporulation of each of our selected *Trichoderma* isolates were explored using the PDA medium.

Czapek’s medium (Appendix A) was used as the basic medium to study the optimum carbon sources (sucrose was replaced with the same amount of glucose, D-fructose, soluble starch, maltose, D-mannitol or no carbon source) and nitrogen sources (sodium nitrate was replaced with the same mass fraction of beef extract, peptone, yeast extract, ammonium sulfate, ammonium dihydrogen phosphate or no nitrogen source) for each of our selected *Trichoderma* isolates.

At 2 DPI, colony diameter was measured to evaluate colony growth. At 10 DPI, conidia production was measured to evaluate sporulation: 10 mL of sterile water was added onto the plate to rinse the colony, while the surface of the colony was gently and thoroughly scraped with an inoculation needle, forming conidia suspension. Then, conidia in the suspension were counted using a blood cell counting plate under the microscope. Three replicates were performed for each experiment.

To test the thermotolerance of our selected *Trichoderma* isolates, a conidia suspension (concentrations of collected Th2, Th3, Th4 were 1.3 × 10^2^ cfu/mL, 4.8 × 10^2^ cfu/mL, 0.6 × 10^2^ cfu/mL, respectively; all adjusted to 0.6 × 10^2^ cfu/mL and then subjected to subsequent treatments) of each isolate was placed in a 1.5 mL centrifuge tube and subjected to a water bath at 40, 45, 50, 55, 60 or 65 °C for 10 min. A total of 200 μL of the processed conidia suspension was then spread on the PDA medium and incubated for two days to test conidia survival.

### 2.7. Statistical Analyses

The MEGA 11 software (Ver. 11.0.13) and the neighbor-joining method were employed for phylogenetic analysis (bootstrap confidence, 1000 replicates, p-distance model). Data were presented as mean ± standard deviation ($\bar{x}$ ± s). Significance was assessed by Student’s *t*-test at *p* < 0.05. Radar charts were generated using the free online tool at https://www.bioinformatics.com.cn/, (accessed on 30 January 2024).

## 3. Results

### 3.1. Trichoderma Strains Were Isolated and Identified

From the compartmentalized *A. lancea* (the rhizome and the fibrous roots) and the associated soil samples, we obtained a total of 63 fungal isolates that exhibited the characteristic colony morphology of *Trichoderma* spp.: the colony had a white, fluffy-looking surface and margin; the colony size expanded rapidly and completely covered the medium in the 9 cm diameter Petri dish within three days post-inoculation (DPI); and the mycelia were fine, looked exquisite and grew expansively.

We further screened for isolates that could rapidly and massively produce conidia by observing the color change and the appearance of concentric circular patterns on the colony. Of the total 63 fungal isolates, we selected 21 that turned yellow to dark green and developed circular patterns on the colony within 7 DPI for voucher preservation (all preserved in the China Academy of Chinese Medical Sciences), identification and further screening. The rest were discarded.

We subjected these 21 isolates to molecular identification by ITS sequencing and obtained clean and clear nucleotide signals for each of them, indicating that all our isolates were pure cultures. BLAST results showed that *T. harzianum* strains accounted for the majority of our vouchers (Table 1).

### 3.2. Screening of Strong Biocontrol Trichoderma Candidates

We screened each of the 21 *Trichoderma* isolates for strains with strong inhibitory effects against Fo and Fs using dual-culture (DC) assays. The inhibition rate of each *Trichoderma* strain on PDA at 10 DPI against Fo or Fs ranged from approximately 50% to 75%. We selected the top three *Trichoderma* strains that most robustly suppressed both Fo and Fs for further experiments. These were three *T*. *harzianum* strains. We named them Th2 (isolated from the rhizome rhizosphere of *A. lancea* of Liaoyang, Liaoning Province), and Th3 and Th4 (isolated from the rhizosphere of fibrous root and rhizome, respectively, of farmed *A. lancea* in Chifeng, Inner Mongolia Autonomous Region). The inhibition rates of Th2, Th3 and Th4 against Fo were 77.04%, 75.89% and 78.62%, respectively, after 10 days of PDA culture. The inhibition rates of the three strains against Fs after 10 days of PDA culture were 78.49%, 77.45% and 76.41%, respectively (Figure 1A–D).

The colonies of Th2, Th3 and Th4 were able to spread further after occupying all the free space on the PDA medium after 10 DPI, gradually reducing the size of the Fo and Fs colonies. The sporulation of Th3 on Fo and Th2 on Fs was obvious after 20 DPI (Figure 1A,C). Microscopic observation 24 h post-inoculation (HPI) showed contact with intersecting, overlapping and intertwining Trichoderma mycelia with/on Fusarium mycelia (Figure 1G–L). These results were all evidence of mycoparasitism of our *Trichoderma* isolates on Fo and Fs, as the majority of *Trichoderma* spp. do to pathogens and other microbes [35,36].

We then investigated the inhibitory effect of volatile organic compounds (VOCs) emitted from Th2, Th3 or Th4 using the dual-culture method of Li et al. [23] (Figure 1E). For 3DPI, the inhibition rates of Th2, Th3 and Th4 on Fo were 46.80%, 46.80% and 35.70%, respectively; their inhibition rates on Fs were 80.00%, 79.50% and 75.70%, respectively (Figure 1F). Our results showed that Th2, Th3 and Th4 were all powerful biocontrol agents with multiple functional mechanisms against root rot pathogens.

### 3.3. Morphological and Phylogenetic Characterization of Th2, Th3 and Th4

On the PDA medium at 28 °C in the dark, the growth rates of Th2, Th3 and Th4 were similar. For Th2, Th3 and Th4, the colony diameter was able to reach 9 cm each and fill the Petri dish in 3 to 4 DPI. Sporulation was observed under the microscope as early as 24 HPI. In 5 to 7 DPI, the conidia clusters were able to form clear concentric circles around the inoculation site (Figure 2A,E,I). Meanwhile, the color of the colony gradually changed from white to yellow-green, and became dark green within the next 24 h.

Under the microscope, the hyphae of Th2, Th3 and Th4 were all transparent and had a smooth surface (Figure 2C,G,K). Many secondary branches were visible on the main dendritic branch. Conidiophores (‘4’ in Figure 2D,H,L) grew from the lateral branches of the mycelium. The structure of the conidiophores was diverse as shown in Table 2 and ‘2’ in Figure 2D,H,L. The conidia were smooth, spherical or ovoid, and translucent to pale green. The conidia of Th2 and Th3 appeared dispersed, while the conidia of Th4 tended to aggregate (Figure 2D(3),H(3),L(3)).

Based on the BLAST results in NCBI, we downloaded the top ten ITS sequences with the highest similarity to Th2, Th3 and Th4. We then performed a phylogenetic analysis based on these ITS sequences. The results suggested that Th2 and Th4 were closely related by chance despite their different origins, while Th3 had a relatively distant connection to Th2 or Th4 (Figure 3). We also performed sequencing and phylogenic analyses of the TEF1 and RPB2 sequences of Th2, Th3 and Th4 (Appendix A). The phylogenic relationships among their TEF1 sequences indicated the same as their ITS (Appendix A).

### 3.4. Th2, Th3 and Th4 Promoted the Medicinal Compound Accumulation of A. lancea Seedlings at the Cost of Plant Growth

First of all, the survival rate of *A. lancea* plantlets inoculated with our selected *Trichoderma* strains was higher than the pathogen-treated group, except for Th4. Most of the surviving *A. lancea* plants in the groups treated with Fo + Fs had necrotic tissue among their fibrous roots, but their rhizomes appeared to be healthy (Figure 4, Appendix A).

We harvested the inoculated *A. lancea* plantlets after growing them in the growth chamber for 70 days. The harvested plants of each experimental group all had a clearly developed rhizome, in contrast to the 30-day-old *A. lancea* plants that had not developed a recognizable rhizome compartment in our previous studies [29,37]. We divided each *A. lancea* plant into three compartments, namely the shoot (the aboveground compartment), the rhizome and the fibrous roots, and measured the weight of each of these compartments. Despite the known growth-promoting role of *T. harzianum* strains [18,21,28] and our expectation, these three *Trichoderma* strains that we deliberately selected did not utterly promote the biomass of *A. lancea* (the Th2, Th3 and Th4 groups in Figure 5).

Most surprisingly, treatment with a mixed pathogen (the Fo + Fs group) significantly boosted the medicinal compound accumulation in the rhizome of surviving *A. lancea* plants (Figure 5). However, the fresh biomass and rhizome dry weight were all lower than in CK (Figure 5). The concentrations of atractylon, atractylodin, hinesol and β-eudesmol were improved to varying degrees in all inoculated groups. It is well known that plants are constantly adjusting and balancing their growth and defense; when one is overly active, the other would have to be relatively slowed down [38]. The four major medicinal compounds of our interest are all secondary metabolites that represent *A. lancea*’s investment in defense rather than growth. Our results showed that Th2, Th3 and Th4 could each influence the balance between defense and growth of *A. lancea*. They could all promote the secondary metabolism of *A. lancea*, but at the cost of suppressing growth.

Compared to the CK group, the atractylon content in the dry rhizome of the Fo + Fs_Th2, Th2_Fo + Fs, Th2, Fo + Fs_Th3, Th3_Fo + Fs, Th3, Fo + Fs_Th4, Th4_Fo + Fs and Th4 groups increased by 13.05, 3.86, 1.52, 6.21, 7.61, 17.22, 5.33, 11.55 and 1.19 times, respectively (Figure 6C,G,K). The atractylodin content in the dry rhizome of the groups Fo + Fs_Th2, Th2_Fo + Fs, Th2, Fo + Fs_Th3, Th3_Fo + Fs, Th3, Fo + Fs_Th4, Th4_Fo + Fs and Th4 increased by 10.63, 3.04, 0.74, 8.00, 9.20, 9.53, 2.23, 8.15 and 1.14 times, respectively (Figure 6D,H,L).

To visualize the growth–defense balance adjusted by our selected *Trichoderma* strains, we generated radar plots showing the fold change of each trait compared with CK (Figure 7). Overall, Th3 had the most optimal effects on the survival rate of *A. lancea* against root rot caused by *Fusarium* spp. and on the accumulation of medicinal compounds in the *A. lancea* rhizome at a relatively low cost of a decrease in rhizome biomass. Our results emphasize the possible selectivity and preference between *A. lancea* and the beneficial *Trichoderma* strain(s).

### 3.5. Establishment of the Optimal Culturing and Sporulation Conditions of Th2, Th3 and Th4

Although of our three selected *Trichoderma* strains only Th3 showed a comprehensive effect on the quality and application potential of the *A*. *lancea* herb, Th2 and Th4 could also promote the accumulation of all four major medicinal compounds in the *A*. *lancea* root. Therefore, we investigated the optimal cultivation and sporulation protocols for all three *T*. *harzianum* strains using a range of culture medium formulas as a reference for fermentation (Figure 8).

As a result, we found that the optimal conditions for the growth of Th2 colonies on the PDA or PSA media, at pH = 6, were in the dark at 25–30 °C. The optimal sporulation conditions for Th2 were on the PSA, CZA, CMA or RBA media, at pH = 5 to 7, 25 °C in the dark; the optimal carbon sources were glucose, soluble starch or D-mannitol, and the optimal nitrogen source was ammonium dihydrogen phosphate (Figure 8).

The optimal conditions for the growth of Th3 colonies were on the PDA, PSA or SDA media at 30 °C, with a pH = 5 to 6. For the sporulation of Th3, the optimal conditions were on the CMA medium, at pH = 6 to 7 at 30 °C in the dark. The most suitable carbon sources for sporulation were sucrose and D-fructose, and the best nitrogen source was ammonium nitrate (Figure 8).

The optimal growth conditions for the Th4 colony were on the PDA medium, at 25 to 30 °C, pH = 6 in the dark. Th4 produced the most spores on the CMA medium, at pH = 6 to 7 at 30 °C; the optimal carbon sources were maltose or D-mannitol; and the optimal nitrogen sources were yeast extract powder or ammonium nitrate (Figure 8).

As a reference for drying *Trichoderma* fermentation products, we tested the thermotolerance of the conidia of Th2, Th3 and Th4. The results showed that the lethal maximum temperature for all three strains was 50 to 55 °C (Table 3), indicating that drying for their fermentation products should be below 50 °C.

## 4. Discussion

In this study, we isolated *Trichoderma* strains from the rhizosphere of the rhizome and the fibrous roots of *A. lancea* samples separately since the chemical composition of the rhizome and the fibrous roots of *A. lancea* is quite different and the literature suggests that the microbiota preference and microbial community structure in the rhizosphere are strongly influenced by the metabolites in the plant root [39]. The roots of medicinal plants such as *A*. *lancea* usually contain a high concentration of certain secondary metabolites, which often have antibiotic effects [40]. Beneficial microbial strains that have growth-promoting effects on crops may not colonize the roots of medicinal plants smoothly and exert the expected functions on medicinal plants. Based on these considerations and concerns, we started to isolate novel *Trichoderma* strains instead of directly using common commercial strains. The identification results of our *Trichoderma* isolates showed no obvious preference between the different underground compartments or geographical origins of *A*. *lancea*. It is possible that the amount of *Trichoderma* isolates we selected for conservation was too small to produce statistical differences. The survival rate of the Th4 group was alarmingly low, indicating an incompatibility between our Jiangsu *A*. *lancea* material and Th4, for which their different geographic origins may be a cause.

In this study, for each selected *Trichoderma* strain, we created two groups for pathogen–*Trichoderma* dual treatments. Theoretically, plant pathogens trigger systemic acquired resistance (SAR), while plant growth-promoting microbes such as *Trichoderma* spp. trigger induced systemic resistance (ISR). The underlying signaling networks and mechanisms of action in plants show some overlap but are nevertheless different [37,38,39]. It has been reported in the literature that plants could respond to *Trichoderma* induction within 24 HPI, at the level of defense-related enzyme activity [41], which represents the activation of ISR. In addition, it has been reported in the literature that plants could activate NO, ROS and certain gene-mediated signaling pathways underlying SAR within 24 HPI [42,43]. We designed the pathogen-first groups (the Fo + Fs_Th groups) to simulate a context in which our *Trichoderma* isolates could control root rot of SAR-active *A*. *lancea* plants. With the Th_Fo + Fs groups, we aimed to simulate a context in which *Trichoderma*-pretreated *A*. *lancea* seedlings with activated ISR grow in soils with a high incidence of *Fusarium* spp. and outbreaks of root rot. Although it is reported in the literature that SAR/ISR is fully activated via multiple pathways after a longer induction period or slightly longer after induction, such as 24 to 72 HPI [44,45,46,47], we were only able to treat our *A*. *lancea* plantlets with the pathogen or *Trichoderma* for 24 h at a time to ensure *A*. *lancea* survival before transplanting into peat soil.

Nevertheless, we obtained different effects between the Fo + Fs_Th group and the Th_Fo + Fs group for each selected *Trichoderma* strain. It was absolutely amazing that some *A*. *lancea* plants treated with the pathogen and *Trichoderma* had both a higher rhizome biomass and higher concentration of medicinal compounds compared to CK, especially the Th3 groups. Compared to the CK group, the Fo + Fs_Th3 and Th3_Fo + Fs groups increased the dry weight of seedling rhizomes by 44.2% and 35.2%, respectively. In the groups that were subjected to a double treatment with Th2 or Th4, the concentration of medicinal compounds in the rhizome was also significantly increased, but at the cost of a greatly reduced biomass of the rhizome. To actually achieve higher herb yields and economic benefits, the survival rate of *A*. *lancea* plants treated with these two treatment groups urgently needs to be improved. Modifying the inoculation techniques or reducing the *Trichoderma* inoculum concentration might help.

The *Trichoderma* products available on the market (biofertilizers in combination with organic material or dried fungal inoculum) are usually labeled as suitable for a wide range of crops, vegetables or fruits. The first aim of *Trichoderma* products is to increase the yield of these crops as a food source. However, it should be noted that the case is completely different for medicinal plants. Due to the known growth–defense balance of plants, the promotion of growth could lead to a decrease in the accumulation of secondary metabolites. In a word, there is probably a balance between yield and quality in cultivated herbs. Certain secondary metabolites are the basis for the clinical effect of medicinal plants. Our study emphasizes the importance of selecting specific biocontrol microbial strains for the cultivation of high-quality medicinal plants.

## Figures and Tables

**Figure 1 jof-10-00351-f001:**
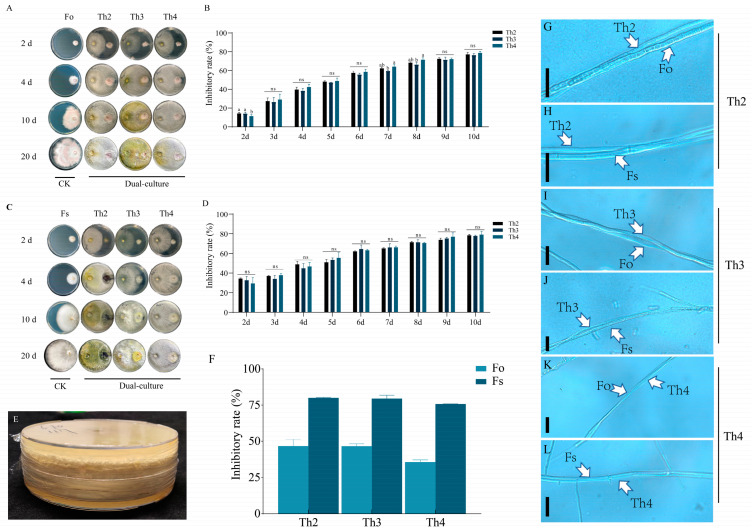
Results of the dual-culture experiments of Th2, Th3 and Th4 with Fo or Fs and microscopic observation (**A**–**D**). d, days of incubation. CK, pathogen culture alone as a control. (**B**,**D**) The inhibition rate of Th2, Th3 and Th4 against the two pathogens after 2–10 days of dual-culture (n = 3); (**E**) an exemplary photo showing the dual-culture method for testing the effects of *Trichoderma* VOC. (**F**) The inhibition rate of the volatile metabolites emitted by Th2, Th3 and Th4 against the two pathogens; (**G**–**L**) microscopic observation of dual-culture mycelia of Th2, Th3 or Th4 with Fo (**G**,**I**,**K**) or Fs (**H**,**J**,**L**) at 24 HPI. Different lowercase letters represent significant differences (*p* < 0.05). Bars: (**G**–**L**), 20 μm.

**Figure 2 jof-10-00351-f002:**
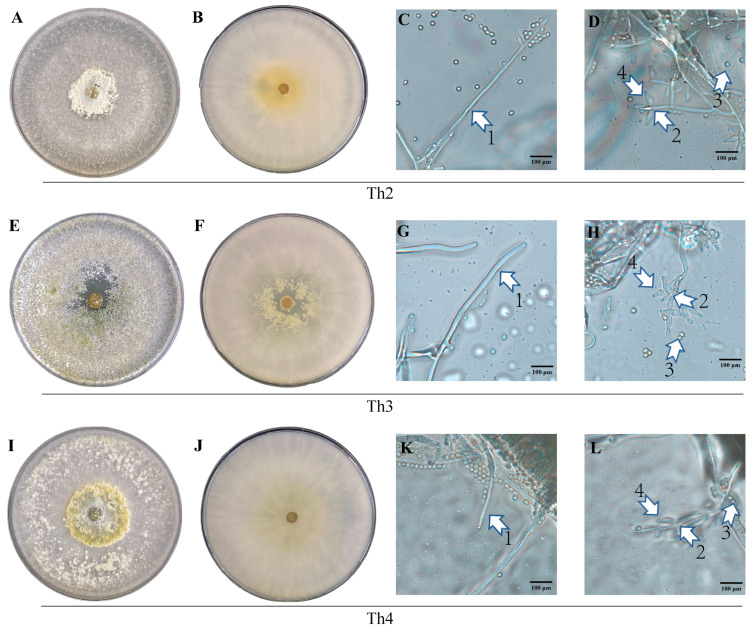
Morphological and molecular identification of Th2, Th3 and Th4. (**A**,**B**,**E**,**F**,**I**,**J**): Th2, Th3 and Th4 colony cultured on PDA medium in a 9 cm diameter petri dish 5 days post-inoculation. (**C**,**D**,**G**,**H**,**K**,**L**): Microscopic morphology of Th2, Th3 and Th4; arrow: 1, mycelium. 2, conidiophores. 3, conidia. 4, phialide. Bars: (**C**,**D**,**G**,**H**,**K**,**L**), 100 μm.

**Figure 3 jof-10-00351-f003:**
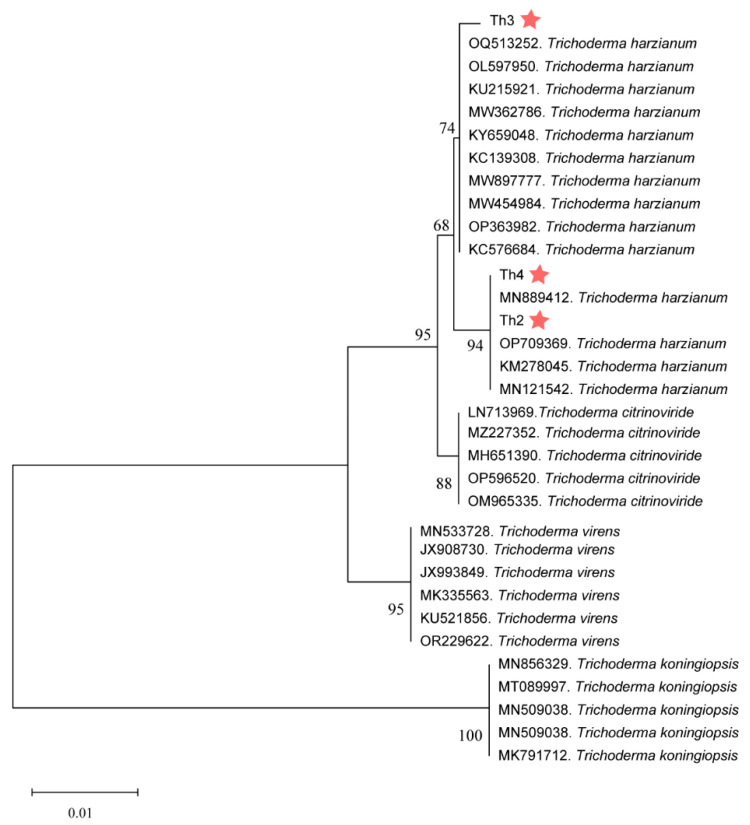
Phylogenetic tree of Th2, Th3, Th4 (marked with red stars) and related strains constructed based on their ITS sequences.

**Figure 4 jof-10-00351-f004:**
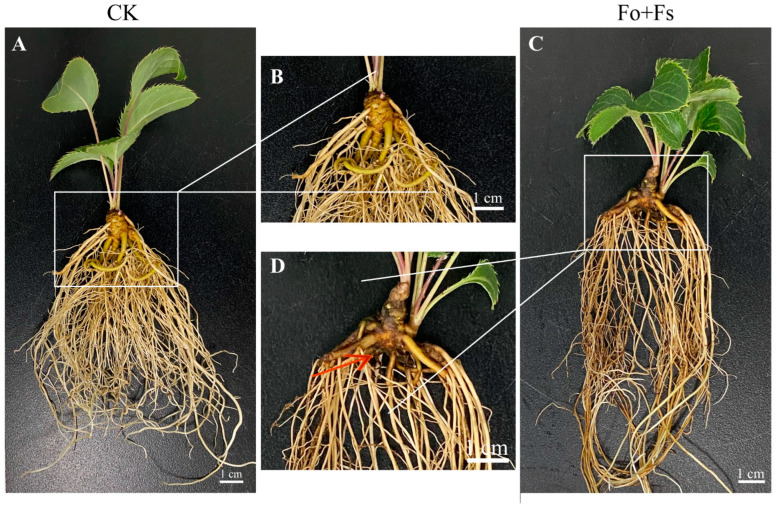
Comparison of phenotypes between the control *A. lancea* plants (**A**,**B**) and those treated with Fo + Fs (**C**,**D**). (**C**,**D**): partial zoom-ups of (**A**) and (**B**), respectively. The red arrow indicates root rot symptom. Bars: (**A**–**D**), 1 cm.

**Figure 5 jof-10-00351-f005:**
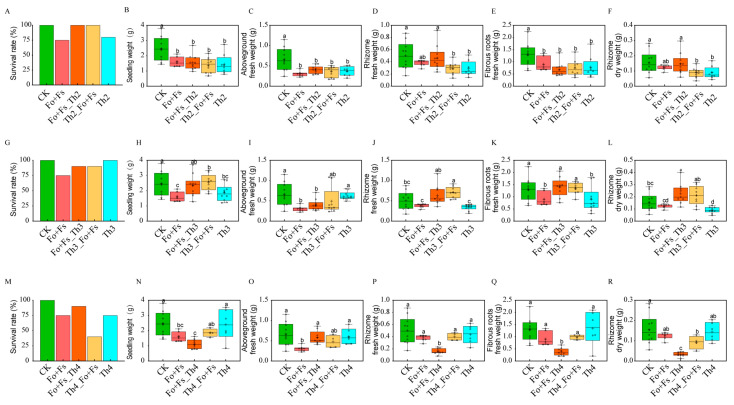
Effects of fungal inoculation with Th2, Th3 and Th4 on *A. lancea* survival and growth. (**A**,**G**,**M**): Survival rate of the inoculated *A. lancea* seedlings. (**B**,**H**,**N**): Seedling weight. (**C**,**I**,**O**): Aboveground fresh weight. (**D**,**J**,**P**): Rhizome fresh weight. (**E**,**K**,**Q**): Fibrous roots’ fresh weight. (**F**,**L**,**R**): Rhizome dry weight. CK, Fo + Fs_Th2, Th2_Fo + Fs, Th3: n = 10; Fo + Fs_Th3, Th3_Fo + Fs, Fo + Fs_Th4: n = 9; Fo + Fs: n = 7; Th4: n = 6; Th4_ Fo + Fs: n = 4. Different lowercase letters represent significant differences (*p* < 0.05). Corresponding data can be found in Appendix A.

**Figure 6 jof-10-00351-f006:**
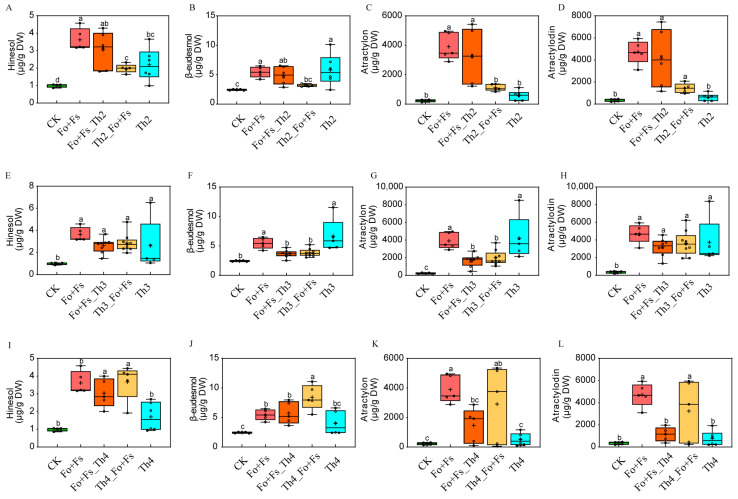
Effects of fungal inoculation with Th2, Th3 and Th4 on the accumulation of the four major medicinal compounds of *A. lancea*. (**A**–**L**): hinesol (**A**,**E**,**I**), β-eudesmol (**B**,**F**,**J**), atractylon (**C**,**G**,**K**) and atractylodin (**D**,**H**,**L**) content in the rhizome. Different lowercase letters represent significant differences (*p* < 0.05). Corresponding data can be found in Appendix A.

**Figure 7 jof-10-00351-f007:**
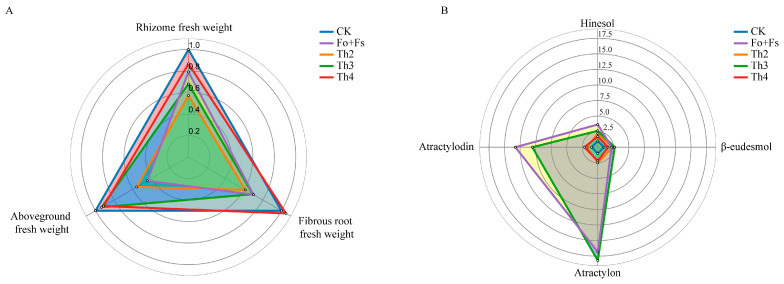
Radar charts showing the fold change of biomass (**A**) and content of medicinal compounds (**B**) in different treatment groups compared to CK. Numeric annotation represents fold change compared to CK.

**Figure 8 jof-10-00351-f008:**
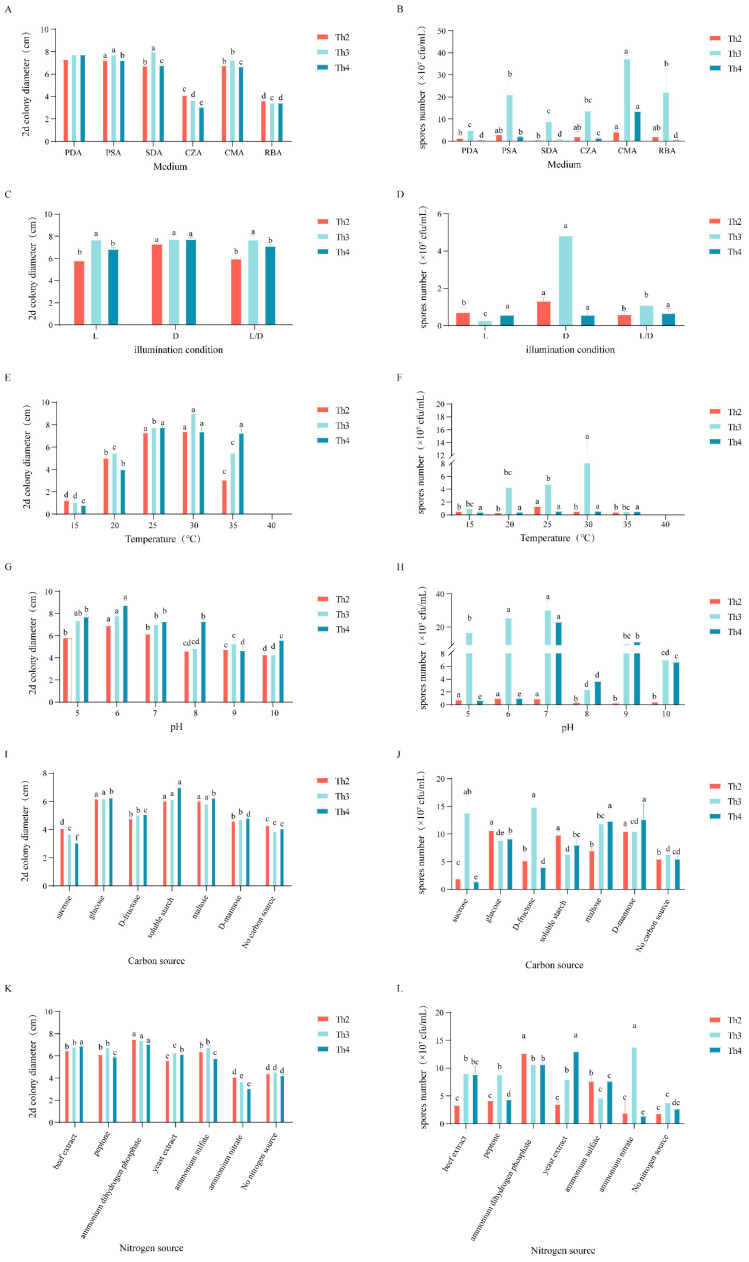
Results of colony growth rate and conidia production under varied culturing regimes (n = 3). (**A**,**B**): Medium; (**C**,**D**): light/dark cycles; (**E**,**F**): temperature; (**G**,**H**): pH; (**I**,**J**): carbon source; (**K**,**L**): nitrogen source. Different lowercase letters represent significant differences (*p* < 0.05). Corresponding data can be found in Appendix A.

**Table 1 jof-10-00351-t001:** Identification result of the *Trichoderma* isolates.

Species Name	Total Isolate Count	Isolate Count by Geographic Origin		Isolate Count by Association to *A. lancea* Compartment	
		Liaoning Province	Inner Mongolia Autonomous Region	Rhizome Rhizosphere	Fibrous Root Rhizosphere
*Trichoderma atroviride*	1	1	0	0	1
*Trichoderma lixii*	1	1	0	0	1
*Trichoderma chlamydosporicum*	3	3	0	1	2
*Trichoderma harzianum*	7	2	5	5	2
*Trichoderma velutinum*	3	1	2	3	0
*Trichoderma koningiopsis*	1	1	0	0	0
*Trichoderma brevicompactum*	4	0	4	3	1
*Trichoderma citrinoviride*	1	0	1	0	1
Sum	21	9	12	12	8

Note: among the total of 21, one isolate was from bulk soil close to the root but not so close as the rhizosphere.

**Table 2 jof-10-00351-t002:** The microscopic appearance of conidiophores’ structure.

Name	Microscopic Morphology of Conidiophores
Th2	*Gliocladium* type, containing 2–4 branches (1–2 cells long), attached to 3–5 phialides or a combination of secondary branches and phialides. Phialides were short with a thin base and an enlarged middle part.
Th3	*Pachybasium* type, with a main axis, short secondary branches forming a nearly right angle with the axis, with a terminal whorl of 1–3 and occasionally 4 phialides. Phialides were short, barrel-shaped, forming large angles between each other.
Th4	*Verticillium* type. Conidiophore had septate base and was dark in color, with 1–3 (occasionally up to 5) slim phialides per whorl at the terminal.

**Table 3 jof-10-00351-t003:** The thermotolerance test results of *Trichoderma* conidia (n = 3).

	T (°C)	40	45	50	55	60	65
*Trichoderma*							
Th2		+	+	+	−	−	−
Th3		+	+	+	−	−	−
Th4		+	+	+	−	−	−

Note: conidia suspension was incubated in hot water bath for 10 min and then spread on PDA medium and cultured to test conidia survival. ‘+’ represents the presence of hyphae and conidia survival; ‘−’ represents no mycelium appearance, conidia were killed by the heat (n = 3).

## Data Availability

Data are contained within the article and Appendix A.

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
