# Peer review of "Allochthonous Trichoderma Isolates Boost Atractylodes lancea Herb Quality at the Cost of Rhizome Growth"

_jof, 2024, doi:10.3390/jof10050351_

Round 1
Reviewer 1 Report
The paper titled "Allochthonous Trichoderma isolates boost Atractylodes lancea herb quality at the cost of rhizome growth" reveals a pertinent contribution to the existing body of knowledge regarding the influence of Trichoderma strains on the root rot resistance, medicinal compound accumulation, and yield of Atractylodes lancea. The study furnishes valuable insights into the potential utility of Trichoderma strains as biocontrol agents and their consequential effects on the growth and defense mechanisms of A. lancea. Notably, the findings elucidate the intricate interplay between defense mechanisms and growth promotion orchestrated by Trichoderma strains, underscoring their capacity to enhance secondary metabolism while concurrently suppressing rhizome growth. Moreover, the study underscores the significance of discerning specific biocontrol microbial strains for cultivating high-quality medicinal plants, thus underscoring the delicate equilibrium between yield and quality in herbal cultivation.
The commendable attributes of the study encompass the adoption of a dual-culture method to delineate the impact of Trichoderma strains on root rot resistance and medicinal compound accumulation in Atractylodes lancea. Further commendation is warranted for the meticulous selection of three Trichoderma strains (Th2, Th3, and Th4) exhibiting robust antagonistic properties against root rot pathogens. Moreover, the exploration of the inhibitory potential of volatile organic compounds (VOCs) emitted by these strains on root rot pathogens stands as a noteworthy endeavor. The study's inquiry into the optimal culturing and sporulation conditions for these strains also merits acclaim, given its pivotal role in facilitating their practical implementation in agricultural contexts.
However, avenues for improvement are discernible, primarily revolving around an enhanced discussion regarding the practical applicability of Trichoderma strains in agriculture and horticulture, encompassing the attendant challenges and limitations, which could augment the study's comprehensiveness and utility.
In summary, the paper presents a significant advancement in understanding the intricate dynamics between Trichoderma strains and Atractylodes lancea, thereby enriching the discourse surrounding biocontrol strategies in herbal cultivation. While commendable in several aspects, opportunities for refinement exist, particularly in extrapolating the study's findings to practical agricultural settings.
The article under review furnishes a comprehensive synthesis of the current state of knowledge within the realm of research, centering on the ramifications of Trichoderma strains on the root rot resistance, medicinal compound accumulation, and rhizome yield of Atractylodes lancea. The research design is deemed apt, with methodological elucidations provided with requisite clarity. The results are articulated cogently, delineating pertinent details, and the ensuing conclusions are buttressed by the empirical findings. Notably, the study engenders valuable insights into the plausible utility of Trichoderma strains as biocontrol agents, whilst shedding light on their nuanced influence on the growth and defense mechanisms of Atractylodes lancea.
Avenues for enhancement encompass a more expansive discussion about the practical deployment of Trichoderma strains in agriculture and horticulture, encompassing a nuanced exploration of challenges and limitations inherent in their application. Although the study alludes to the utilization of Trichoderma strains in agricultural and horticultural contexts, a paucity of specific details regarding their practical integration into Atractylodes lancea cultivation is apparent.
Furthermore, a more comprehensive discussion surrounding the challenges and limitations entailed in the utilization of Trichoderma strains in agriculture and horticulture would be salutary. Embracing topics such as requisite conditions and the propensity for pathogen resistance development would foster a more holistic understanding of the feasibility of deploying Trichoderma strains in Atractylodes lancea farming.
In summation, the article's contribution emerges as pertinent to the extant corpus of knowledge concerning the effects of Trichoderma strains on Atractylodes lancea, whilst furnishing invaluable insights into the putative utility of these strains as biocontrol agents. However, opportunities for refinement are discernible for enriching discussions surrounding their practical applicability in agricultural settings.
The text is well-written and easy to understand. However, there are a few minor improvements that could be made to enhance the clarity and precision of the language.
Line 26: "The RBA plates and were then placed in an incubator at 28°C in dark, and closely observed by naked eye every day."
Suggested correction: "The RBA plates were then placed in an incubator at 28°C in the dark and observed daily by naked eye."
Line 29: "Morphology of novel isolates was observed every day, only those showed characteristics of Trichoderma spp. (fast colony growth, fluffy mycelium structure, gradual color change of colony to yellow, green, then dark green) were subjected to further culturing and identification."
Suggested correction: "The morphology of the novel isolates was observed every day, and only those that exhibited characteristics of Trichoderma spp. (fast colony growth, fluffy mycelium structure, gradual color change of colony to yellow, green, then dark green) were subjected to further culturing and identification."
Line 32: "Identification of potential Trichoderma isolates were performed via microscopic observation and sequencing of the ITS (internally transcribed spacer) sequence with the 1TS1 5′⁃TCCGTAGGTGAACCTGCGG⁃3′ and ITS4 5′-TCCTCCGCTTATTGA- TATGC-3′ primer pair."
Suggested correction: "The identification of potential Trichoderma isolates was performed via microscopic observation and sequencing of the ITS (internally transcribed spacer) sequence using the 1TS1 5′⁃TCCGTAGGTGAACCTGCGG⁃3′ and ITS4 5′-TCCTCCGCTTATTGA- TATGC-3′ primer pair."
Line 43: "Fungal vouchers and the pathogens (Fo and Fs) were cryogenically preserved in 5- mm diameter round pieces of PDA culture immersed in 50 % (v/v) glycerol at −80 °C at the Chinese Academy of Chinese Medical Sciences."
Suggested correction: "Fungal vouchers and the pathogens (Fo and Fs) were cryogenically preserved in 5-mm diameter round pieces of PDA culture immersed in 50% (v/v) glycerol at −80°C at the Chinese Academy of Chinese Medical Sciences."
Line 152: "The biocontrol potentials of the Trichoderma isolates against Fo and Fs were evaluated via dual-culture (DC) experiments on PDA in 9-cm-diameter petri dishes following Wang et al."
Suggested correction: "The biocontrol potentials of the Trichoderma isolates against Fo and Fs were evaluated using the dual-culture (DC) method on PDA in 9-cm-diameter petri dishes, following the method of Wang et al."
Line 157: "Meanwhile, dual-culture experiments following the method of Ma et al. were performed to test the inhibitory effects of the volatile organic compounds of our Trichoderma isolates (Figure 1E)."
Suggested correction: "Meanwhile, dual-culture experiments following the method of Ma et al. were performed to test the inhibitory effects of the volatile organic compounds of our Trichoderma isolates (Figure 1E)."
Line 160: "An Fo or Fs plate sealed with a blank PDA plate was taken as the control (CK)."
Suggested correction: "An Fo or Fs plate was taken as the control (CK), with a blank PDA plate used as the control."
Line 162: "Inhibitory rate (IR, %) = (DCK - DDC) / DCK × 100 %."
Suggested correction: "Inhibitory rate (IR, %) = (DCK - DDC) / DCK × 100%."
Line 163: "2.4 Inoculation of A. lancea seedlings."
Suggested correction: "2.4 Inoculation of A. lancea seedlings."
Line 169: "Liquid fungal culture in PDB was used as inoculum."
Suggested correction: "Liquid fungal culture in PDB was used as inoculum."
Line 170: "Liquid Fo and Fs culture were mixed to form the pathogen inoculum, in which the concentration of Fo and Fs was 2.5×107 cfu/mL each."
Suggested correction: "Liquid Fo and Fs culture were mixed to form the pathogen inoculum, with the concentration of Fo and Fs being 2.5×107 cfu/mL each."
Line 172: "Fungal concentration of the liquid Th2, Th3 and Th4 culture was 6.0×106 cfu/mL, 1.0×107 cfu/mL and 1.0×107 cfu/mL, respectively."
Suggested correction: "The fungal concentration of the liquid Th2, Th3 and Th4 culture was 6.0×106 cfu/mL, 1.0×107 cfu/mL and 1.0×107 cfu/mL, respectively."
Line 173: "They were all adjusted to 6.0×106 cfu/mL using sterile water and subsequently used as the Trichoderma inocula."
Suggested correction: "They were all adjusted to 6.0×106 cfu/mL using sterile water and subsequently used as the Trichoderma inocula."
Line 174: "Inoculation was performed via placing the hardened A. lancea plantlets in tissue-culture bottles, root immersed in sterile water and subjected to a 48-hour hardening process in the growth cabinet."
Suggested correction: "Inoculation was performed by placing the hardened A. lancea plantlets in
Reviewer 2 Report
Dear authors,
The topic of the manuscript is very interesting and above all important for medical plant production. The manuscript is well-written and easy to follow. Material and methods are sufficiently explained, results are well presented but Figures 5 and 6 should be enlarged because numbers and letters near axis are too small.
My major remark would be that Discussion section needs to be improved, especially considering results from Figures 5 and 6. I miss the explanation how happened that Fo+Fs treatment had higher results than a control and Trichoderma treatments in Figure 6? Also, how do you explain Figure 5M - survival rate of Th4_Fo+Fs was lower that Fo+Fs treatment? Also, Figure 5A showed that survival rate of Fo+Fs was similar to Th2 treatment. You mentioned in line 375 that "Th2 and Th4 could also produce undesirable results" - did you notice some kind of necrosis or have some observations from previous studies? Please, explain or specify that.
My additional question would be - why did you use only ITS primer for Trichoderma identification? Usually, combination of ITS and EF or RPB primers gives more reliable results.
- Figure 2 - please define which photo belongs to which isolate (Th2 - A, B, C i D?)
- Figures 5 and 6 should be enlarged
- lines 347 and 338 - A. lancea should be italic
- line 323: knwon => known
